# Exploring Advanced Therapies for Primary Biliary Cholangitis: Insights from the Gut Microbiota–Bile Acid–Immunity Network

**DOI:** 10.3390/ijms25084321

**Published:** 2024-04-13

**Authors:** Ziqi Guo, Kun He, Ke Pang, Daiyu Yang, Chengzhen Lyu, Haifeng Xu, Dong Wu

**Affiliations:** 1Peking Union Medical College Hospital, Chinese Academy of Medical Sciences and Peking Union Medical College, Beijing 100730, China; guozq18@student.pumc.edu.cn (Z.G.); pangk19@student.pumc.edu.cn (K.P.); yangdy20@student.pumc.edu.cn (D.Y.); 2Department of Gastroenterology, State Key Laboratory of Complex Severe and Rare Diseases, Peking Union Medical College Hospital, Chinese Academy of Medical Sciences & Peking Union Medical College, Beijing 100730, China; hk6290418@163.com (K.H.); chengzhenlyu@outlook.com (C.L.); 3Department of Liver Surgery, Peking Union Medical College Hospital, Chinese Academy of Medical Sciences & Peking Union Medical College, Beijing 100730, China

**Keywords:** primary biliary cholangitis, gut microbiota, bile acid

## Abstract

Primary biliary cholangitis (PBC) is a cholestatic liver disease characterized by immune-mediated injury to small bile ducts. Although PBC is an autoimmune disease, the effectiveness of conventional immunosuppressive therapy is disappointing. Nearly 40% of PBC patients do not respond to the first-line drug UDCA. Without appropriate intervention, PBC patients eventually progress to liver cirrhosis and even death. There is an urgent need to develop new therapies. The gut–liver axis emphasizes the interconnection between the gut and the liver, and evidence is increasing that gut microbiota and bile acids play an important role in the pathogenesis of cholestatic diseases. Dysbiosis of gut microbiota, imbalance of bile acids, and immune-mediated bile duct injury constitute the triad of pathophysiology in PBC. Autoimmune cholangitis has the potential to be improved through immune system modulation. Considering the failure of conventional immunotherapies and the involvement of gut microbiota and bile acids in the pathogenesis, targeting immune factors associated with them, such as bile acid receptors, microbial-derived molecules, and related specific immune cells, may offer breakthroughs. Understanding the gut microbiota–bile acid network and related immune dysfunctions in PBC provides a new perspective on therapeutic strategies. Therefore, we summarize the latest advances in research of gut microbiota and bile acids in PBC and, for the first time, explore the possibility of related immune factors as novel immunotherapy targets. This article discusses potential therapeutic approaches focusing on regulating gut microbiota, maintaining bile acid homeostasis, their interactions, and related immune factors.

## 1. Introduction

Primary biliary cholangitis (PBC), previously known as primary biliary cirrhosis, is a chronic immune-mediated cholestatic disease that occurs predominantly in women, with a female-to-male ratio of approximately 4–6:1 [1]. PBC is an autoimmune disease with autoantibodies anti-mitochondrial antibodies (AMA) in 90 to 95% of patients, which also serves as a critical diagnostic marker [2]. The etiology of PBC is multifactorial. It is believed to be linked to genetic and environmental factors, but the pathogenesis of the disease remains unclear [3]. Currently, there are limited therapeutic agents for PBC, with two guideline-recommended drugs: ursodeoxycholic acid (UDCA) and obeticholic acid (OCA) [4]. Nevertheless, around 30–40% of patients treated with UDCA do not respond adequately, and OCA is reportedly associated with a higher incidence of adverse events [2,5].

PBC is characterized by the destruction of intrahepatic small bile ducts and cholestasis. Bile acids play a crucial role in the pathophysiology of cholestatic diseases; their accumulation in the liver can lead to hepatocyte inflammation and oxidative stress, resulting in hepatotoxicity [6]. The gut–liver axis describes the close functional link between the gut and the liver. The microbiota in the human intestine acts as a regulator of health, which plays a crucial role in coordinating a variety of homeostatic processes such as metabolism and immune function. However, gut dysbiosis can lead to intestinal barrier dysfunction, allowing gut microbiota, its derivatives, and associated activated immune cells to reach the liver through the portal circulation and exert their effects on the liver [7]. Gut microbiota contributes to increased intestinal permeability and inflammation during cholestasis, actively promoting liver injury [8]. Growing evidence reveals the association between the gut microbiota and liver diseases. In PBC patients, dysbiosis of gut microbiota was observed and showed partial restoration following UDCA treatment [9]. In addition, there exist complex interactions between gut microbiota and bile acids. Gut microbiota provides key enzymes for the conversion of primary bile acids to secondary bile acids, changing the composition of the bile acid pool. Bile acids affect the abundance and composition of gut microbiota due to their antibacterial activity and also impact the intestinal barrier function through related receptors [10]. Recent studies have also revealed the interaction of PBC with gut microbiota and bile acids, describing the dysbiosis of gut microbiota and the alteration of bile acid composition in PBC patients [9,11,12,13,14,15,16,17,18]. These provide new perspectives for exploring potential therapeutic approaches for PBC.

Although the pathogenesis of PBC involves immune-mediated bile duct inflammation, immunosuppressive therapy has largely been shown to be ineffective as monotherapy for PBC [19]. This may be due to the predominance of bile acid-mediated liver damage at this stage [20]. Given the significant role of bile acids in PBC pathogenesis and their interaction with gut microbiota, immune factors associated with bile acids and gut microbiota are directions worth exploring and may serve as potential intervention targets. This review discusses potential therapeutic approaches of PBC in terms of regulating gut microbiota, maintaining bile acid homeostasis, their interaction, and associated immune factors and provides recent advancements in some drugs under investigation.

## 2. Methods

We conducted searches on both PubMed and Google Scholar in February 2024 using the following keywords: “primary biliary cholangitis” or “primary biliary cirrhosis” or “cholestatic liver disease” or “autoimmune liver disease”, “gut microbiota” or “intestinal microbiota” or “gut microbiome” or “intestinal microbiome”, “bile acid” and “immunity” or “immune” or “immune response” or “immunotherapy”. K.P., D.Y. and C.L. conducted the initial literature search, Z.G., K.H., K.P., D.Y. and C.L. reviewed relevant studies for appropriate inclusion. Furthermore, we reviewed relevant additional articles from the references of retrieved papers.

## 3. Gut Microbiota and Bile Acids in PBC

### 3.1. Dysbiosis of Gut Microbiota and Impaired Intestinal Barrier in PBC

With recent advancements in the study of the composition and function of human microbiota, the gut microbiota is increasingly recognized for its role in promoting health and its association with the development or maintenance of different gastrointestinal and non-gastrointestinal diseases [21]. The interaction between gut microbiota and PBC is gaining recognition, with numerous studies showing alterations in the diversity and composition of gut microbiota in PBC. In an animal experiment, the dnTGFβRII mice exhibited reduced proportions of *S24-7*, *Ruminococcaceae*, *Rikenellaceae*, and *Porphyromonadaceae*, while showing increased levels of *Lachnospiraceae* and *Bacteroidaceae* compared to the controls [22]. Moreover, the T-cell-mediated cholangitis in mice was alleviated after antibiotic treatment [22]. NOD.c3c4 mice serve as an additional PBC model, with research indicating milder liver disease in mice raised under germ-free conditions [23], reflecting the potential impact of microbiota on disease progression. Alterations in the gut microbiota have also been identified in PBC patients, mostly through studies using fecal samples, with limited research utilizing mucosal biopsy samples [9,11,12,13,14,17,18]. The primary findings from these studies are summarized in Table 1. Additionally, there is also research focused on the salivary microbiota to investigate the potential pathogenic link between oral microbial communities and PBC [17,24]. Most studies have consistently reported diminished diversity in the gut microbiota of PBC patients when compared to healthy controls [9,13,14,18]. An increased abundance of certain pathogenic bacteria alongside a decrease in beneficial bacteria have been observed. *Klebsiella*, found in the gut microbiota of PSC patients, has been implicated in disrupting the epithelial barrier, leading to bacterial translocation and eliciting inflammatory responses in the liver [25]. Its enrichment has also been noted in the fecal samples of PBC patients [9,11,12]. In addition, *Clostridia*, comprising numerous beneficial species, was found to be diminished [13]. The level of total bilirubin, which serves as a marker for late-stage disease in PBC, shows a correlation with the gut microbiome profile [15]. This discovery reveals the relationship between gut microbiota dysbiosis and the prognosis of PBC, suggesting a potential contribution of gut dysbiosis in disease progression.

Gut microbiota is affected by multiple complex factors, such as diet and ethnicity [26], and the changes in gut microbiota in PBC patients from different regions exhibit slight differences [14]. Currently, most research on the gut microbiota in PBC was conducted in China or Japan, suggesting potential regional specificity in the findings obtained. Furthermore, differences in disease status among the PBC patients enrolled in different studies might also contribute to divergent outcomes. Additionally, these findings are primarily observational, indicating a link between gut microbiota and PBC, which potentially impacts disease prognosis. However, the precise mechanisms by which gut microbiota exert its effects on PBC are unclear. The causal relationship between gut microbiota and disease remains undetermined. A recent study found three previously unrecognized bacteria taxa causally related to PBC [27]. The class *Coriobacteriia* and order *Coriobacteriales* are associated with the risk of PBC, whereas class *Deltaproteobacteria* acts as a protective factor. Moreover, dysbiosis in gut microbiota can affect the intestinal barrier and alter microbe-derived metabolites, consequently disrupting immune homeostasis [28].

Evidence has indicated the gut barrier dysfunction in PBC, which acts as a gateway for bacteria and microbiota-derived metabolites to enter the portal circulation and reach the liver, activating the immune system and inducing or worsening hepatic inflammation. In a mouse model of PBC exhibiting gut barrier dysfunction (dnTGFβRIITLR2^−/−^ mice), the increased gut permeability is demonstrated to promote gut bacterial translocation to the liver and exacerbate hepatic T-cell-mediated cholangitis [22]. Studies revealed increased gastrointestinal permeability in PBC patients with findings of increased sucrose excretion [29], elevated serum LPS levels, and higher antibody titers of lipoteichoic acid (LTA) [30,31]. Secretory IgAs (SIgA), antimicrobial peptides and mucus collectively serve as a chemistry barrier, forming the primary defense against microbiota and pathogens [28]. Defects in in situ IgA secretion in the intestinal epithelium and widening of the cell-to-cell junction were observed in patients with PBC compared to controls [32]. Gut barrier dysfunction leads to bacteria translocation, disrupting the balance between host immunity and microbiota.

### 3.2. Imbalance of Bile Acids in PBC

Bile acids not only modulate the composition of intestinal microbes but also act as important metabolites of microbes, playing a pivotal role in gut–liver axis communication [33]. In cholestatic diseases such as PBC and PSC, excess bile acids are cytotoxic and can cause damage to liver cells. Primary bile acids, chenodeoxycholic acid (CDCA), and cholic acid (CA) are synthesized in the liver and then conjugated with glycine or taurine to form conjugated hydrophilic bile acids. Under the action of specific enzymes produced by gut microbiota, primary bile acids are converted into secondary bile acids such as deoxycholic acid (DCA), lithocholic acid (LCA), and ursodeoxycholic acid (UDCA). Bile acids interact bidirectionally with gut microbiota, shaping the gut microbiota and influencing intestinal barrier function, and gut microbiota in turn participate in the metabolism of bile acids. Various bile acid transport proteins such as bile salt export protein (BSEP) and apical sodium-dependent bile acid transporter (ASBT) are present in hepatocytes and intestinal epithelial cells to facilitate the secretion and reabsorption of bile acids, completing the enterohepatic circulation (Figure 1) [3,10].

At low concentrations, bile acids function as crucial signaling molecules regulating metabolism and the immune system; as the concentration rises, they will cause apoptosis, induce inflammation, and yield necrosis [34]. Excessive accumulation of bile acids in the bile ducts in cholestatic diseases is considered cytotoxic, leading to hepatocyte damage. Bile acids at high cellular concentrations may act as danger-associated molecular patterns (DAMPs) leading to the activation of the NLRP3 inflammasome [35]. Anion exchange 2 (AE2) is a Cl^−^-HCO3^−^ anion exchanger, crucial for maintaining a biliary HCO3^−^ “umbrella” that protects biliary epithelial cells (BECs), while reduced AE2 in BECs was observed in PBC patients [34,36]. Hydrophobic bile acids, notably glycochenodeoxycholic acid (GCDCA), have been observed to decrease the expression of AE2 in BECs [37]. Additionally, elevated levels of GCDCA have been detected in the serum of UDCA treatment-naive PBC patients [12]. These findings suggest a potential involvement of bile acids in PBC. Furthermore, gut microbiota produces bile acid metabolism-related enzymes such as bile salt hydrolase (BSH), hydroxysteroid dehydrogenase, and 7α-dehydroxylase, which participate in the synthesis of secondary bile acids, altering hydrophobicity and toxicity of bile acids. Dysbiosis in the gut microbiota consequently leads to alterations in the bile acid pool. Significant differences in the bile acid profiles in both serum and feces were found between UDCA treatment-naive PBC and controls, with a lower total secondary/primary ratio and an increased conjugated/unconjugated ratio of bile acids in naive PBC patients compared with controls [12]. The reduced levels of secondary bile acids were associated with a decline in the abundance of *Faecalibacterium* and *Oscillospira* genera, which were enriched in the controls [12]. Due to the altered composition of bile acids in PBC, bile acids may also serve as potential biomarkers for the disease. 12-dehydrocholic acid (12-DHCA), a secondary bile acid derived from CA, was shown to be specifically elevated in the urine of responders in PBC patients undergoing UDCA therapy, indicating its potential as a biomarker for treatment response [38].

### 3.3. Immune Response Mediated by Gut Microbiota and Bile Acids

Microbial dysbiosis and disruption of the intestinal barrier lead to the breakdown of intestinal homeostasis, allowing pathogenic microorganisms to pass through the gut into the circulation. Increased microbial pathogen associated molecular patterns (PAMP) such as lipopolysaccharides (LPS) in the portal circulation bind to Toll-like receptors (TLR) on BECs, triggering the pro-inflammatory NF-κB pathway and inducing the release of cytokines like CX3CL1, consequently leading to cellular damage [3]. It is worth mentioning that LPS activates TLR4, modulating the transforming growth factor-beta (TGF-β) signal through the MyD88-NF-kappaB-dependent pathway, which is associated with hepatic inflammation and fibrosis [39].

Molecular mimicry is a mechanism of autoimmunity caused by foreign substances such as in the gastrointestinal microbiome. It occurs when foreign proteins or peptides closely resemble self-peptides, which may activate autoreactive T or B cells that attack host cells [40]. Recurrent urinary tract infection is strongly associated with PBC and has been indicated as a risk factor for the development of PBC, which may be due to molecular mimicry between the PDC-E2 (target of AMA) epitopes in human and *Escherichia coli* [41]. *Novosphingobium aromaticivorans* is another bacterium that may be connected to PBC by molecular mimicry. The abundance of *Novosphingobium* (also called *Sphingomonas*) was also observed to be enriched in the ileal mucosa of patients with PBC [18]. The immune complex formed by AMA and the antigen PDC-E2 can activate local and regional dendritic cells, triggering innate immune responses in PBC, resulting in cellular disruption, and finally causing progressive bile duct destruction [42].

The various metabolites produced by gut microbiota also influence the host’s immune response. Short-chain fatty acids (SCFAs), such as acetate, propionate, and butyrate, are mainly produced from dietary fiber by gut microbial fermentation [43]. SCFAs are generally thought to be beneficial to the host, which provide nutrients to the intestinal epithelium, strengthen gut barrier function, enhance the antimicrobial activity of macrophages, promote the generation of anti-inflammatory regulatory T-cells, and participate in attenuating the hyperinflammatory responses caused by pathogenic bacteria [44,45,46,47]. However, the findings regarding SCFAs in PBC patients are controversial. Some studies have reported a significantly reduced presence of butyrate-producing beneficial bacteria in the UDCA non-responder group and an enrichment of SCFA-producing bacteria in the superior remission group after cholestyramine treatment in PBC patients [13,48]. Nevertheless, no differences in fecal or serum SCFAs were found between UDCA responder and non-responder groups in another study [38]. Lammert et al. noted that PBC patients with fibrosis exhibited higher levels of total fecal SCFAs and acetate compared to those without fibrosis, which suggests that SCFAs may be associated with fibrosis progression in PBC [16]. Whether SCFAs play different roles in various stages of disease progression remains to be further investigated.

The disruption in bile acid balance also affects the immune regulation of the body. At high cellular concentrations, bile acids may activate the NLRP3 inflammasome. Hydrophobic primary bile acid CDCA participates in cholestatic liver injury by triggering macrophage NLRP3 inflammasome activation and the release of the pro-inflammatory cytokine IL-1β [49]. GCDCA affects AE2 expression in BEC by inducing reactive oxygen species (ROS), and decreased expression of AE2 enhances the production of CXCL10, IL-8, and IL-6 in BEC, facilitating the migration of immune cells to BEC [37]. CA inhibits the self-renewal of intestinal stem cells and obstructs the restoration of intestinal epithelial barrier function, exacerbating intestinal inflammatory responses and barrier impairment [50]. In addition, CA accumulation in the liver promotes hepatic stellate cells activation through early growth response protein 3 (EGR3), driving liver fibrosis [51]. Hydrophobic secondary bile acids, such as DCA and LCA, demonstrate immunosuppressive properties by activating G protein-coupled bile acid receptor 1 (GPBAR1, also called TGR5), which suppresses LPS-induced cytokine expression in Kupffer cells, and ameliorating liver inflammation by inhibiting the nuclear factor κB (NF-κB) pathway [52,53]. Evidence in patients with ulcerative colitis has also shown that the deficiency of secondary bile acids induced by gut dysbiosis exacerbates intestinal inflammation [54]. Moreover, specific secondary bile acids, two derivatives of LCA known as 3-oxoLCA and isoalloLCA, have been demonstrated to modulate adaptive immunity by inhibiting Th17 cell differentiation and increasing regulatory T cell (Treg) differentiation [55]. Farnesoid X receptor (FXR) and TGR5 are the two most discussed bile acid receptors that are expressed in the gut and liver, as well as in various immune cells such as macrophages, dendritic cells, and NKT cells, and are involved in the control of immune response. They are negative regulators of the NLPR3 inflammasome [35].

## 4. Present and Potential Therapy

The relationship between bile acid, gut microbiota, and primary biliary cholangitis highlights targeting them as a potential strategy in PBC treatment. Apart from regulating gut microbiota and bile acid, considering the involvement of immune-mediated bile duct injuries in PBC, targeting related immune factors is also a promising direction.

### 4.1. Targeting Gut Microbiota

#### 4.1.1. Antibiotics

In autoimmune cholestatic disorders, such as PBC and PSC, alterations in gut microbiota have been documented, manifesting as an increase in the abundance of pathogenic bacteria and a reduction in gut microbial diversity. In a mouse model of PBC, a milder liver phenotype was observed when NOD.c3c4 mice were treated with antibiotics [23]. Similarly, another animal study demonstrated the alleviation of cholangitis through antibiotic treatment [22]. Several studies have been undertaken to evaluate the use of antibiotics in the treatment of PSC; nonetheless, the outcomes remain controversial. Vancomycin appeared promising as an antibiotic treatment for PSC. Multiple clinical trials have documented improvements in liver blood tests and reductions in Mayo risk scores with vancomycin [56,57]. However, a retrospective study showed similar outcome metrics among the vancomycin-treated group, the UDCA-treated group, and the untreated group [58]. Dissimilar to PSC, there is a notable lack of clinical trials investigating antibiotic therapy in patients with PBC. Currently, the only antibiotic drug utilized in clinical therapy is rifampicin, a second-line drug recommended for the management of itching. However, the efficacy of rifampicin may not be attributable to modification of the gut microbiome, as a recent study found no significant differences in the fecal bacterial composition among PBC patients with pruritus, those without pruritus, and healthy controls [59]. The effectiveness of using antibiotics to alter the gut microbiota for treating PBC requires further investigation.

#### 4.1.2. Probiotics, Prebiotics, and Synbiotics

Probiotics, living microorganisms ingested for their health benefits, might hold therapeutic promise for cholestatic diseases by modulating intestinal microbiota and regulating bile acid metabolism [60]. In mice with bile duct ligation and multidrug resistance protein 2 knockouts (Mdr2^−/−^), the effects of the probiotic *Lactobacillus rhamnosus* GG (LGG) on hepatic bile acid synthesis, liver damage, and fibrosis were examined [61]. The study showed that probiotic LGG protects the liver by enhancing bile acid excretion and reducing hepatic bile acid synthesis through upregulation of the intestinal farnesoid X receptor-fibroblast growth factor 15 (FXR-FGF15) signaling pathway. Probiotic LGG treatment could modify the gut microbiota and enhance microbes with BSH activity, promoting the deconjugation of bile acids and thus enhancing their excretion. Furthermore, another probiotic *Lactobacillus plantarum* Lp2 has demonstrated hepatoprotective properties against liver injury in mice [62,63]. Currently, the only trial to assess the efficacy of probiotics in PBC patients with inadequate response to UDCA is ongoing, with an unknown status (NCT03521297).

The efficacy of prebiotics and synbiotics in non-alcoholic fatty liver disease (NAFLD) and non-alcoholic steatohepatitis (NASH) has been assessed in several clinical trials [64,65,66]. At present, there are no studies investigating the use of prebiotics or synbiotics in PBC and PSC.

#### 4.1.3. Fecal Microbiota Transplantation

Fecal microbiota transplantation (FMT) is a therapeutic approach designed to transfer gut microbes from a healthy donor to a recipient to modify the composition of the intestinal microbiome for treating intestinal and extraintestinal diseases. FMT has been recommended for the management of recurrent *Clostridium difficile* infection [67]. Concurrently, numerous trials are underway to investigate its potential therapeutic indications, such as inflammatory bowel disease, autism spectrum disorder, metabolic syndrome, and liver diseases [68,69]. In an open-label pilot clinical trial, 10 patients with PSC-IBD received FMT from a single healthy donor, and a more than 50% decrease in ALP levels was observed in 30% of them [70]. This study confirmed the short-term safety and efficacy of FMT as a prospective therapeutic intervention for PSC. Nevertheless, the utilization of FMT in PBC patients or animal models has not been investigated by any current research. Given the potential involvement of gut microbiota in PBC and the established safety of FMT in other liver diseases, the exploration of FMT as a potential treatment for PBC is worthy of implementation.

#### 4.1.4. Bacteriophages

The human gut contains a large number of microbes, among which viral particles far outnumber bacteria, but little is known about the intestinal virome in patients with liver disease. Bacteriophages are viruses that specifically target certain bacteria, and they have the ability to selectively eradicate pathogenic bacteria while minimally impacting the overall gut environment. A recent preclinical study confirmed that bacteriophage targeting of gut bacteria can attenuate alcoholic liver disease [71]. Bacteriophage therapy was also explored and evaluated in PSC. A trial showed that a lytic phage cocktail specifically targeted and suppressed *Klebsiella* spp. could attenuate hepatobiliary inflammation and slow fibrosis progression in PSC mouse models [72]. While studies on bacteriophage therapy in PBC are lacking, deeper research and comprehension of the pathogenic mechanisms linking PBC and intestinal microbiota might establish bacteriophages as a novel targeted therapy.

### 4.2. Targeting Bile Acids Homeostasis

Currently, the treatment of PBC primarily focuses on the regulation of bile acids; the approved agent UDCA and OCA remain the first treatment option for patients with PBC. Additionally, there are several potential treatments under research. Table 2 provides an overview of present and potential therapies targeting bile acids in PBC.

#### 4.2.1. UDCA

UDCA is an endogenous secondary bile acid produced in humans and most other species by gut bacteria metabolism. As a hydrophilic bile acid with low cytotoxicity, UDCA is increasingly used for the treatment of cholestatic diseases and is recommended by the FDA as a first-line therapeutic drug for PBC [89,90]. UDCA can not only improve biochemical liver tests but also delay disease progression and prolong the survival of PBC patients. In an international multicenter study of 3902 patients with PBC, the application of UDCA has been proven to prolong liver transplantation-free survival irrespective of disease stage and biochemical response [91]. UDCA exerts a protective effect on the liver through multi-aspect mechanisms, including modification of the bile acid pool, stimulation of hepatobiliary bile acid excretion, protection of cholangiocytes against bile acid-induced cytotoxicity and apoptosis, and potential immunomodulatory effects [92,93,94,95]. At different stages of cholestasis, UDCA may exert different mechanisms of action.

During UDCA treatment, exogenous UDCA enriches the pool of hydrophilic bile acids, replacing the endogenous hydrophobic toxic bile acids that accumulate during cholestasis [95]. Whether the enrichment of UDCA in the bile pool affects the gut microbiota is an interesting topic worthy of discussion. A prospective study revealed that differences in the abundance of PBC-associated bacteria between the UDCA-untreated PBC patients and healthy controls were partially reversed after 6 months of UDCA treatment [9]. *Haemophilus* spp., *Streptococcus* spp., and *Pseudomonas* spp. decreased after UDCA treatment, while *Bacteroidetes* spp., *Sutterella* spp., and *Oscillospira* spp. expanded [9]. The alteration of gut microbiota in PBC might result from the direct effect of the UDCA intervention mechanism, or just secondary changes due to the improvement in cholestasis following UDCA treatment. In addition, the taurine-metabolizing bacteria *Bilophila* spp. significantly increased in patients with PBC after UDCA treatment [12]. Another study explored the relationship between biochemical response to UDCA and the gut microbiota composition in patients with PBC, finding a notably lower abundance of the genus *Faecalibacterium* in the UDCA non-responder group [13]. The genus *Faecalibacterium* could potentially act as a predictor of PBC prognosis, and the mechanism of it and PBC deserves further investigation.

#### 4.2.2. FXR-FGF19 Axis

In clinical contexts, FXR is closely associated with cholestasis development and is a key therapeutic target for cholestasis and other liver diseases. FXR functions as a bile acid receptor with pleiotropic effects, including regulation of the absorption, synthesis, transport, and excretion of bile acids. Expressed in the liver and small intestine, FXR is an important regulator of bile acid homeostasis. Its activation inhibits bile acid synthesis by suppressing genes encoding key enzymes in this process. Activation of intestinal FXR promotes the production of FGF19 (FGF15 in mice), which circulates to the liver through the portal vein, binds to FGFR4 receptor and co-receptor β-klotho on the surface of hepatocytes, and suppresses the transcription of the *CYP7A1/Cyp7a1* gene (encoding the key rate-limiting enzyme CYP7A1 in bile acid synthesis) [96]. Activation of FXR in the liver induces the transcription of small heterodimer chaperones (SHP), repressing the transcription of *CYP8B1/Cyp8b1*, while exerting a minor influence on suppressing the expression of *CYP7A1/Cyp7a1* [97,98,99]. Moreover, FXR is involved in the enterohepatic circulation of bile acids by regulating transporters; its activation upregulates the expression of receptors related to bile acid secretion like BSEP and OSTα/β and inhibits bile acid uptake-related receptors like ASBT and NTCP [100]. FXR is also crucial in the crosstalk between gut microbiota and bile acids [10]. The gut microbiota exerts a negative regulatory effect on bile acid synthesis via the FXR–FGF19/15 pathway. A study observed that gut microbiota regulated bile acid metabolism by alleviating FXR inhibition in the ileum [101]. Conversely, bile acids can also modulate the composition of the gut microbiota through FXR. Studies revealed that FXR inhibited bacterial overgrowth and mucosal injury in the ileum of mice subjected to bile duct ligation, thereby decreasing bacterial translocation, which suggests that FXR agonists may prevent epithelial injury and bacterial translocation in patients with impaired bile flow [102,103]. This demonstrates the protective role of FXR in the gut–liver axis.

Endogenous bile acids are natural ligands of FXR, with varying degrees of FXR activation as follows: CDCA > DCA > LCA > CA > all conjugated bile acids [104]. OCA is a semisynthetic bile acid analogue derived from CDCA. It functions as a highly selective agonist of FXR, exhibiting 100-fold higher potency than endogenous bile acids, and it is currently the only agent recommended as a second-line treatment for PBC patients with UDCA tolerance and/or non-response to UDCA by the FDA [105]. A double-blind, placebo-controlled study revealed that OCA increased the transport of bile acids from hepatocytes to bile ducts, consequently reducing the hepatic retention time of potentially toxic bile acids by approximately one-third [106]. Changes in the intestinal microbiota were observed in both healthy humans and mice after OCA treatment [107]. Of note, OCA enhanced gut barrier function and reduced bacterial translocation by upregulating tight-junction protein expression in cholestatic rats [103]. Although OCA demonstrates beneficial effects in cholestatic diseases, pruritus side effects have also been reported in several clinical trials [5,108]. In addition to the approved OCA, several non-steroidal FXR agonists are currently under investigation, such as tropifexor (LJN452), cilofexor, and EDP-305. The safety and efficacy of tropifexor were evaluated in a phase II, double-blind, placebo-controlled clinical trial involving PBC patients, which showed improvement in cholestatic markers in the tropifexor group compared to placebo [73]. The safety, tolerability, and efficacy of cilofexor and EDP-305 were evaluated in trials NCT02943447 and NCT03394924, respectively.

According to the FXR-FGF19 axis mentioned above, FGF19 represents another novel treatment target for PBC. Aldafermin (NGM282), an FGF19 analogue without pro-tumor activity, improved ALP and transaminase levels compared to a placebo in a clinical trial and was confirmed to be safe in PBC patients [74]. These results pave the way for further research to evaluate the clinical benefits of FGF19 analogues in PBC.

#### 4.2.3. PPAR Agonists

Peroxisome proliferator-activated receptors (PPAR) are members of the nuclear hormone receptors family, including three isoforms: PPARα, PPARβ/δ, and PPARγ. PPAR activation contributes to the regulation of bile acid synthesis, bile excretory function, and immune response abnormality [109,110], and it is also considered to be a potential target. PPAR agonists are a hot spot in the development of new therapeutic drugs for PBC, and several drugs have entered clinical phase 2 or 3 research stages. Fibrates, such as bezafibrate and fenofibrate, are agonists of PPAR and have been examined in PBC patients. Several trials have demonstrated that bezafibrate, either used alone or in combination with UDCA, can ameliorate liver biochemistry in PBC patients [76,77]. Bezafibrate has been used as a second-line treatment for patients who respond incompletely to UDCA in Japan. A large retrospective cohort study conducted in Japan demonstrated an association between the combined use of bezafibrate and UDCA and the improved prognosis in PBC [75]. Adding fenofibrate to treat incomplete responders to UDCA in PBC patients can improve ALP levels [79,80]. For treatment-naive patients, their combination significantly enhances the biochemical response rate [78], which suggests that earlier starting points for adding fenofibrate therapy may improve benefits. Other PPAR ligands such as seladelpar (MBX-8025), elafibranor (GFT-505), and saroglitazar are also currently undergoing investigation. Seladelpar is a selective PPARδ agonist, with reported safety and efficacy in improving liver biochemical response and alleviating pruritus in PBC patients [81,82,83,84]. In a recently published phase 3 clinical trial, seladelpar demonstrated significant improvements compared to placebo in terms of biochemical response, normalization of alkaline phosphatase, and alleviation of itching [111]. Given seladelpar’s excellent performance, in February 2024, the FDA accepted the new drug application for seladelpar from CymaBay and approved it for the treatment of primary biliary cholangitis in adults. Elafibranor, a dual PPARα/δ agonist, has demonstrated significantly greater improvement in cholestasis-related biochemical markers compared to the control group, with a similar incidence of adverse reactions observed in both treatment groups [85]. Saroglitazar has shown the ability to rapidly and continuously improve ALP levels in PBC patients. However, a higher incidence of transaminase elevation was observed at higher doses [86]. A clinical trial using lower doses of 1 and 2 mg of saroglitazar is currently ongoing (NCT05133336).

#### 4.2.4. ASBT Inhibitors

The ASBT protein located in the terminal ileum is involved in maintaining bile acid homeostasis. The expression of ASBT is downregulated in patients with cholestatic liver diseases [112], possibly representing an adaptive change in the body. ABST inhibitors, alternatively known as ileal bile acid transporter (IBAT) inhibitors, block the enterohepatic circulation by suppressing bile acid reabsorption in the intestine and thus decrease bile acid load. In a mouse model of PSC, cholestatic liver and bile duct injury were alleviated after ASBT inhibitor treatment [113]. Several trials have been conducted on ASBT inhibitors in PBC, although most of these trials have concentrated on improving pruritus. Linerixibat (GSK2330672) is a selective inhibitor of IBAT; its function in ameliorating pruritus was investigated in clinical trials, showing positive outcomes [87,88]. However, it is worth noting that the increased accumulation of bile acids in the gut would lead to diarrhea and may have secondary effects on the gut microbiota. Correspondingly, diarrhea was the most frequently reported adverse event linked to ASBT inhibitor therapy in clinical trials [87,88]. The clinical application of ASBT needs further experimental exploration and a trial of another agent, volixibat, is ongoing (NCT05050136).

### 4.3. Targeting Immune Factors Related to Gut Microbiota and Bile Acid

#### 4.3.1. TGR5

TGR5 acts as a receptor for endogenous bile acids, and its activation negatively regulates inflammation. Activation of TGR5 reduces the activity of the NF-κB pathway and shifts the phenotype of macrophages from pro-inflammatory to anti-inflammatory (M1 to M2) [114]. NKT cells are an important component of the liver immune system, and TGR5 was shown to be involved in the regulation of NKT phenotype. Deletion of the TGR5 gene in mice shifts the phenotype of NKT cells towards pro-inflammatory NKT1 cells that produce IFN-γ. Conversely, the application of TGR5 agonists induces the differentiation of NKT cells towards regulatory NKT10 cells that produce IL-10 [115]. Furthermore, activation of TGR5 in cholangiocytes promotes the secretion of chloride and bicarbonate, protecting hepatocytes from the toxicity of bile acids [115]. In terms of hemodynamics, TGR5 may reduce portal vein pressure by regulating the secretion of vasodilators and vasoconstrictors [116]. Considering the multiple physiological roles of TGR5, TGR5 could potentially become a therapeutic target for liver diseases. Currently, TGR5 agonists have not been used in clinical applications. Several TGR5 agonists, including INT-777, BAR501, and the dual agonist of FXR and TGR5, INT-767, were tested in mice [116].

#### 4.3.2. Microbial-Derived Molecules

SCFAs affect intestinal barrier function and host immune response, playing an important role in promoting intestinal homeostasis. SCFAs affect immune responses through various cells, including intestinal epithelial cells, dendritic cells, and macrophages, and are important metabolites that promote the differentiation of naive T lymphocytes into anti-inflammatory regulatory T cells (Treg) [117]. The understanding of the relationship between SCFAs and bile acid remains relatively limited. Sheng et al. demonstrated that microbial transplantation with butyrate-deficient feces exacerbated liver inflammation and bile acid dysregulation in Western diet-fed FXR knockout mice, while these effects were reversed by butyrate supplementation [118]. The regulatory mechanisms of SCFAs on bile acid metabolism are controversial. Dietary supplementation with SCFA promoted bile acid excretion and upregulated *Cyp7a1* gene expression in male Syrian hamsters, while in another study, it was found that SCFA had no effect on *Cyp7a1* mRNA expression [119,120]. Considering the beneficial effects of SCFAs on gut microbiota homeostasis and their potential involvement in regulating bile acid metabolism, SCFAs have the potential to become a therapeutic target. *Clostridium* metabolite p-Cresol sulfate (PCS) can suppress the immune response of Kuffer cells in the liver, shifting them from M1 to M2, alleviating inflammation in PBC [121]. Dietary interventions or supplementing PCS may be effective clinical strategies for PBC treatment.

#### 4.3.3. Immune Cells and Signaling

In patients with PBC, liver-infiltrating T cells are localized to the biliary epithelial cells and make significant contributions to the pathogenesis, rendering them a potential therapeutic target. However, current targeted T cell therapies for cholestatic liver disease have not yielded satisfactory outcomes. Glaser et al. discovered that CD8^+^ T cells influence bile acid synthesis and metabolism through the pro-inflammatory cytokines TNF and IFN-γ, resulting in decreased levels of toxic unconjugated bile acids [122]. The existence of such a protective mechanism may lead to the poor efficacy of non-selective targeted T cell therapy for immune-mediated biliary diseases. Huang et al. identified significantly expanded and hyperactivated CD103^+^ T_RM_ cells in the liver of PBC patients, which are a major component of PDC-E2 specific autoreactive CD8^+^ T cells and exhibit cytotoxicity against cholangiocytes [123]. NUDT1 regulates the hepatic accumulation of CD103^+^ T_RM_ cells in PBC, promoting the long-term survival of CD103^+^ T_RM_ cells by resisting DNA damage through the PARP1-TGFβ-Smad axis [123]. Targeted inhibition of CD103^+^ T_RM_ cells in the liver may alleviate immune-mediated biliary damage in PBC patients. Wnt/β-catenin signaling is involved in various aspects of liver physiology, such as bile acid metabolism and liver fibrosis. Inhibition of β-catenin expression can reduce the activation of the β-catenin/FXR complex, thereby reducing total bile acids and alleviating cholestatic injury [124]. OP-724 is a CBP-β-catenin antagonist whose tolerability was well demonstrated in patients with advanced PBC in a phase 1 clinical trial [125].

### 4.4. Targeting the Interaction of Bile Acid and Gut Microbiota

The gut microbiota–bile acid network plays an important role in the pathogenesis of PBC; the interaction of them is worth considering as a therapeutic direction. The crosstalk between the two in liver diseases has been described in detail elsewhere [10]. The gut microbiota participates in bile acid metabolism and regulates bile acid synthesis through the FXR-FGF15/19 pathway via negative feedback. Bile acids can also affect the gut microbiota through FXR. The activation of FXR can prevent the excessive proliferation of intestinal microorganisms and play a positive role in the protection of the intestinal barrier, promoting gut homeostasis [10]. FXR plays a crucial role in the interaction between gut microbiota and bile acids, which further confirms its potential as a therapeutic target. However, due to the widespread expression of FXR throughout the body, clinical studies have shown that systemic FXR agonists often exhibit significant adverse reactions. Intestine-restricted FXR agonists, which are being evaluated in metabolic liver diseases [116], may be a worthwhile option to explore. FXR agonists targeting specific tissues or cells can achieve efficacy while avoiding potential side effects, which deserve further study in the future.

As mentioned earlier, the conversion of primary to secondary bile acids is reduced in PBC patients, and the levels of secondary bile acids are associated with specific microbial populations [12]. Further exploration of the link between gut microbiota and bile acids in disease states will pave the way for precise manipulation of gut microbiota and their mediated bile acid metabolism. In addition, given the bidirectional interaction between the two, combined therapies that manipulate gut microbiota and regulate bile acid homeostasis may be effective approaches.

## 5. Discussion

Understanding the mechanisms of PBC aids in exploring novel therapeutic targets. Given the significant role of bile acid-mediated inflammation in pathogenesis, targeting bile acids emerges as a highly promising approach. Most bile acid-targeting drugs have progressed into Phase 2 and Phase 3 clinical trials, yielding encouraging results. Among them, the PPARδ agonist seladelpar holds promise as a new recommended medication. Clinical data reveal an association between gut microbiota and PBC, paving the way for gut–liver axis modulation as a novel therapeutic concept. The efficacy of modulating the gut microbiota to improve liver biochemistry and hepatoprotection has been preliminarily demonstrated in preclinical trials and other autoimmune liver diseases. Additionally, microbiota transplantation shows promise in preventing advanced liver fibrosis [126]. More clinical trials are needed to further validate the safety and efficacy of microbiota modulation approaches in the treatment of PBC. Therapies based on gut microbiota and bile acid-related immune factors are currently in the theoretical and preclinical stages. Further investigating how changes in gut microbiota lead to downstream immune reactions and the effects of microbial metabolites including bile acids could provide new insights into treatments.

The epigenetic changes, particularly alterations in small non-coding RNA (miRNA) expression, provide new molecular insights into pathogenesis and treatment in PBC. MiRNAs play a critical role in the development of many diseases, and a single miRNA can regulate multiple genes and pathways. For instance, miR-129-5p influences various diseases ranging from digestive tumors to neurodegenerative disorders by modulating different genes and pathways including WNT/β-catenin and PI3K/AKT/mTOR [127]. The dysregulation of specific miRNAs has also been observed in patients with PBC. MiRNAs modulate immune responses, hepatocyte apoptosis, bile acid metabolism, and biliary fibrosis by inhibiting multiple signaling pathways [128]. In cholestatic mice, the overexpression of miR-210 in the liver leads to bile acid metabolism defection and hepatic inflammation by inhibiting mixed-lineage leukemia-4 (MLL4) (FXR transcriptional coactivator) [129]. Future in-depth research on miRNA may aid in developing novel targeted therapeutics.

## 6. Conclusions

This review discusses the pathogenesis and treatment strategies of PBC from the perspective of gut microbiota, bile acids, and related immune functions, which provides a novel viewpoint for enhancing comprehension of PBC. Clinical data demonstrate an association between gut microbiota and bile acids with PBC, providing a theoretical basis for treatment. Currently, the treatment of PBC primarily focuses on bile acids. In addition to the approved drugs UDCA and OCA, clinical trials have shown promising results with FXR and PPAR agonists. Therapeutic approaches targeting gut microbiota and related immune factors are still in the preclinical stage. Further clinical trials are needed to validate their efficacy and safety. Bile acids, gut microbiota, and immunity are interdependent in the pathophysiology of PBC, and combination therapy targeting related pathways holds promise for the future.

## Figures and Tables

**Figure 1 ijms-25-04321-f001:**
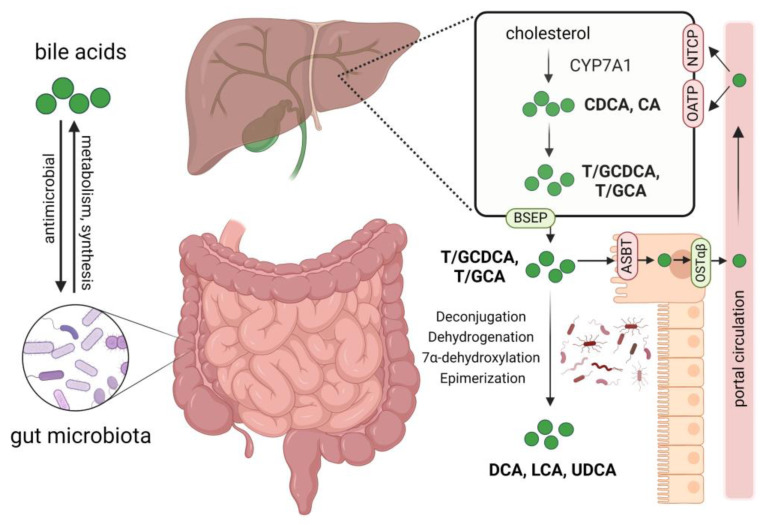
Interactions between bile acids and gut microbiota, bile acid metabolism, and circulation. Gut microbiota participates in the synthesis and metabolism of bile acids, impacting the composition of the bile acid pool. Bile acids affect the abundance and composition of gut microbiota through their antimicrobial activity. Primary bile acids chenodeoxycholic acid (CDCA) and cholic acid (CA) are synthesized from cholesterol in the liver; they then bind to glycine or taurine to form conjugated hydrophilic bile acids. Through the bile salt export protein (BSEP), these bile acids are secreted into the bile and then released into the duodenum; approximately 95% of them are reabsorbed mainly by enterocyte apical sodium-dependent bile acid transporter (ASBT) and effluxed into the portal circulation by the organic solute transporter α and β heterodimer (OSTα/β), finally taken up by hepatocytes via Na^+^-dependent taurocholate cotransport peptide (NTCP) and organic anion-transporters (OATP). The unabsorbed bile acids enter the colon and interact with gut microbiota, forming secondary bile acids such as deoxycholic acid (DCA), lithocholic acid (LCA), and ursodeoxycholic acid (UDCA) by deconjugation, dehydrogenation, 7α-dehydroxylation, and epimerization of primary bile acids. Created by Biorender.com (accessed on 9 March 2024).

**Table 1 ijms-25-04321-t001:** Studies of gut microbiota in PBC.

Studies (Published Time)	Country	Participant (Number)	Sample	Method	Main Findings
Lv et al. (2016) [11]	China	PBC (42) vs. HC (30)	stool	16S rRNA	PBC vs. HC: γ-Proteobacteria↑, *Enterobacteriaceae*↑, *Neisseriaceae*↑, *Spirochaetaceae*↑, *Veillonella*↑, *Streptococcus*↑, *Klebsiella*↑, *Actinobacillus pleuropneumoniae*↑, *Anaeroglobus geminatus*↑, *Enterobacter asburiae*↑, *Haemophilus parainfluenzae*↑, *Megasphaera micronuciformis*↑, *Paraprevotella clara*↑; Acidobacteria↓, *Lachnobacterium* sp. ↓, *Bacteroides eggerthii*↓, *Ruminococcus bromii*↓; potential biomarkers: *Streptococcus* sp. and *Veillonella* sp.
Chen et al. (2020) [12]	China	PBC (79) vs. HC (114)	stool	16S rRNA	PBC vs. HC: *f-Enterobacteriaceae*↑, *Prevotella*↑, *Veillonella*↑, *Fusobacterium*↑, *Haemophilus*↑, *Prevotella*↑, *Streptococcus*↑, *f-Clostridiaceae*↑, *Pseudomonas*↑, *Citrobacter*↑, *Lactobacillus*↑, *Salmonella*↑, *Clostridium*↑, *Klebsiella*↑, *Sneathia*↑; *f-Mogibacteriaceae*↓, *Blautia*↓, *f-Christensenellaceae*↓, *Butyricimonas*↓, *Akkermansia*↓, *Odoribacter*↓, *Dialister*↓, *f-Rikenellaceae*↓, *Oscillospira*↓, *f-S24-7*↓, *Faecalibacterium*↓, *Phascolarctobacterium*↓, *o-Clostridiales*↓, *Sutterella*↓, *f-Barnesiellaceae*↓, *Bacteroides*↓
Furukawa et al. (2020) [13]	Japan	PBC (76) vs. HC (23); UDCA non-responder (30) vs. responder (43)	stool	16S rRNA	PBC vs. HC: diversity↓; *Streptococcus*↑, *Lactobacillus*↑, *Bifidobacterium*↑, *Enterococcus*↑; *Lachnospiraceae*↓, *Ruminococcaceae*↓, *Clostridia*↓; UDCA non-responder vs. responder: *Faecalibacterium*↓
Tang et al. (2018) [9]	China	PBC (60) vs. HC (80); PBC before and after UDCA treatment (37)	stool	16S rRNA	PBC vs. HC: diversity↓ *Haemophilus*↑, *Veillonella*↑, *Clostridium*↑, *Lactobacillus*↑, *Streptococcus*↑, *Pseudomonas*↑, *Klebsiella*↑, an unknown genus in the family of *Enterobacteriaceae*↑; *Bacteroidetes* spp.↓, *Sutterella*↓, *Oscillospira*↓, *Faecalibacterium*↓; PBC after UDCA vs. before UDCA: *Haemophilus* spp.↓, *Streptococcus* spp.↓, *Pseudomonas* spp.↓; *Bacteroidetes* spp.↑, *Sutterella* spp.↑, *Oscillospira* spp.↑
Zhou et al. (2023) [14]	China	PBC (25) vs. HC (25)	stool	16S rRNA	PBC vs. HC: diversity↓; *Acidimicrobiia*↑, *Yersiniaceae*↑, *Serratia*↑, *ucg_010*↑; *Faecalibacterium*↓, *Ruminococcaceae*↓, *Sutterellaceae*↓, *Oscillospiraceae*↓, *Parasutterella*↓, *Clostridia*↓, *Coprococcus*↓, *Christensenellaceae*↓; PBC with anti-gp210-positive vs. anti-gp210-negative: *Oscillospiraceae*↓
Han et al. (2022) [15]	China	PBC TB+ (20) vs. TB− * (27)	stool	16S rRNA	PBC TB+ vs. TB−: diversity↓; *Proteobacteria*↑; *Firmicutes*↓, *Bacteroidetes*↓, *Actinobacteria*↓, Saccharibacteria↓, *Gemmiger*↓, *Blautia*↓, *Anaerostipes*↓, *Coprococcus*↓, Holdemania×
Lammert et al. (2021) [16]	USA	PBC with non-advanced fibrosis (15) vs. advanced fibrosis ** (8)	stool	16S rRNA	PBC with advanced fibrosis vs. non-advanced fibrosis: diversity↓; *Weissella*↑
Abe et al. (2018) [17]	Japan	PBC (39) vs. HC (17)	stool; saliva	16S rDNA	PBC vs. HC (stool): *Lactobacillales*↑; *Clostridium subcluster XIVa*↓; PBC vs. HC (saliva): *Veillonella*↑, *Eubacterium*↑; *Fusobacterium*↓
Kitahata et al. (2021) [18]	Japan	PBC (34) vs. HC (21)	ileal mucosa	16S rRNA	PBC vs. HC: diversity↓; *Sphingomonas*↑, *Pseudomonas*↑, *Methylobacterium*↑, *Carnobacterium*↑, *Acinetobacter*↑, *Curvibacter*↑, an unknown genus *Clostridiaceae*↑; *Leptotrichia*↓, *Morganella*↓, *Lautropia*↓, *Mogibacterium*↓, *Atopobium*↓, *Bulleidia*↓, *Eikenella*↓, *Paludibacter*↓, an unknown genus belonging to the class *TM7_3*↓, and an unknown genus of the family F16↓

↑: increase; ↓: decrease; ×: disappear; PBC: primary biliary cholangitis; HC: health control; UDCA: ursodeoxycholic acid; TB: total bilirubin; *: TB+ indicates TB > 1 × upper limit of the normal range (ULN); TB− indicates TB ≤ 1 × ULN; **: non-advanced fibrosis (≤F2 fibrosis); advanced fibrosis (≥F3 fibrosis).

**Table 2 ijms-25-04321-t002:** Present and potential therapies targeting bile acids in PBC.

Agents	Mechanism	Clinical Status/Outcomes
Approved treatments		
UDCA	Alter BA pool	First-line therapeutic drug
OCA	FXR agonist	Second-line therapeutic drug
Potential treatments		
Tropifexor (LJN452)	FXR agonist	Reduced GGT and ALP levels, the most frequent adverse event was pruritus [73];
Cilofexor	FXR agonist	Phase 2 (NCT02943447)
EDP-305	FXR agonist	Phase 2 (NCT03394924)
Aldafermin (NGM282)	FGF19 agonist	Decreased ALP, GGT, and serum transaminase levels, along with a reduction of C4 and total bile acid levels [74]
Bezafibrate	Pan-PPAR agonist	Combination therapy (with UDCA) improved liver biochemistries, GLOBE and UK-PBC scores and long-term prognosis [75,76,77]; Phase 3 (NCT04751188) Phase 2 (NCT04594694) Phase 2 (NCT05239468)
Fenofibrate	PPARα agonist	Combination therapy (with UDCA) reduced serum ALP levels; improved GLOBE and UK-PBC scores [78,79,80]; Phase 3 (NCT05751967) Phase 2/3 (NCT05749822)
Seladelpar	PPARδ agonist	Improved liver biochemistry and pruritus, decreased C4 and serum bile acids concentration, appeared safe and well tolerated [81,82,83,84]; Phase 3 (NCT06051617) Phase 3 (NCT06060665) Newly approved by FDA
Elafibranor	PPARα + PPARδ agonist	Greater improvements in relevant biochemical indicators of cholestasis, such as ALP and total bilirubin levels [85]; Phase 2 (NCT05627362) Phase 3 (NCT06016842)
Saroglitazar	PPARα + PPARγ agonist	Rapid and sustained improvements in ALP, but a higher incidence of elevated liver enzymes was observed with the 4 mg dose [86]; Phase 2b/3 (NCT05133336)
Linerixibat (GSK2330672)	ASBT inhibitor	Improved pruritus and decreased serum bile acids concentration, the most common adverse event was diarrhea [87,88]; Phase 3 (NCT04950127)
Volixibat	ASBT inhibitor	Phase 2 (NCT05050136)

PBC: primary biliary cholangitis; UDCA: ursodeoxycholic acid; OCA: obeticholic acid; BA: bile acid; FXR: farnesoid X receptor; GGT: γ-glutamyl transferase; ALP: alkaline phosphatase; FGF19: fibroblast growth factor 19; C4: 7α-hydroxy-4-cholesten-3-one (a marker of bile acid synthesis); PPAR: peroxisome proliferator-activated receptors; ASBT: apical sodium-dependent bile acid transporter; GLOBE score and UK-PBC score: two prognostic scores for primary biliary cholangitis.

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
