# Peer review of "Exploring Advanced Therapies for Primary Biliary Cholangitis: Insights from the Gut Microbiota–Bile Acid–Immunity Network"

_ijms, 2024, doi:10.3390/ijms25084321_

Round 1

Reviewer 1 Report

Comments and Suggestions for Authors

The review " Exploring Advanced Therapies for Primary Biliary Cholangitis: Insights from the Gut Microbiota-Bile Acid-Immunity Network" is a true trailblazer in the study of the microbiota and brings forth exceptionally important advancements. However, there are several recommendations that would help enhance the quality of the article:

1.       The introduction section should be expanded. Please provide more information regarding the microbiota.

2. The Discussion section needs to be added. It would greatly help to incorporate information regarding treatment - specifically, the potential benefits of microbiota transfer in microbiome regulation - for example: https://doi.org/10.3390/biomedicines11112930 - where you can also find information related to tumor pathology, but there are other sources to draw from as well. Additionally, you could discuss microRNAs in digestive tumor pathology and beyond. For example: https://doi.org/10.3390/biomedicines11072058. Today, microRNAs provide valuable insights, both through their interaction with specific genes and through the identification of patterns depending on the underlying pathology.

3.      The conclusions need to better incorporate the information from the study.

4.      You can add a section about the materials and methods related to the study selection process.

Comments on the Quality of English Language

Minor editing of English language required.

Reviewer 2 Report

Comments and Suggestions for Authors

In their review, the authors provide an overview of the current literature on the latest advances in research into the gut microbiota and bile acids in primary biliary cholangitis. This article discusses potential therapeutic approaches focusing on the regulation of the gut microbiota, maintenance of bile acid homeostasis, their interactions and associated immune factors.

The review is interesting, well conceived and written, and relevant. The manuscript provides a clear rationale for recent advances in the study of the gut microbiota and bile acids in primary biliary cholangitis and explores for the first time the potential of associated immune factors as novel immunotherapy targets. The literature is relevant and up to date.

I have no objections to the main content of the review, it is really well done. My comments and suggestions for improvement are as follows:

1. The abstract is short and should be expanded. The authors should add a few lines on immune factors that could be considered as new immunotherapy targets, as this is a main topic of this article.

2. As this is a review article, authors should add a paragraph about the literature search, stating the keywords used for the analysis, the databases searched, the number of papers identified, the number of papers excluded, the exact date they conducted the literature search, and the initials of the authors involved in this process.

3. Tables – All abbreviations should be explained in the legend of a table. Please revise.

4. The authors indicate that this work was supported by the Beijing Natural Science Foundation. It is unclear why the authors required support as this is a review article. There were no expenses for this study. Please clarify.

Round 2

Reviewer 1 Report

Comments and Suggestions for Authors

The authors have successfully responded to each recommendation individually.